# Involving men in cervical cancer prevention; a qualitative enquiry into male perspectives on screening and HPV vaccination in Mid-Western Uganda

**Marlieke de Fouw**[1]*, **Yaël Stroeken**[1], **Ben Niwagaba**[2], **Mwalimu Musheshe**[2], **John Tusiime**[2], **Isingoma Sadayo**[2], **Ria Reis**[3,4,5], **Alexander Arnold Willem Peters**[1], **Jogchum Jan Beltman**[1]

1 Department of Gynaecology, Leiden University Medical Centre, Leiden, The Netherlands, 2 Department of Technologies for Rural Transformation, African Rural University, Kagadi, Uganda, 3 Department of Public Health and Primary Care, Leiden University Medical Center, Leiden, The Netherlands, 4 Department of Anthropology, University of Amsterdam, Amsterdam, The Netherlands, 5 Children's Institute, University of Cape Town, Cape Town, South Africa

* marlieke@femalecancerfoundation.org

**Data Availability Statement:** There are restrictions imposed by the ethical review board of the Uganda Cancer Institute on sharing the minimal data for

## Abstract

### Introduction

Evidence-based preventive strategies for cervical cancer in low-resource setting have been developed, but implementation is challenged, and uptake remains low. Women and girls experience social and economic barriers to attend screening and human papillomavirus (HPV) vaccination programs. Male support has been proven successful in uptake of other reproductive healthcare services. This qualitative study with focus groups aimed to understand the perspectives of males on cervical cancer screening and HPV vaccination in Western-Uganda This knowledge could be integrated into awareness activities to increase the attendance of cervical cancer screening and HPV vaccination programs.

### Materials and methods

Focus group discussions were conducted with men aged 25 to 60 years, who were married and/or had daughters, in Kagadi district, Mid-Western Uganda. All interviews were transcribed verbatim and thematically analyzed using an inductive approach.

### Results

Eleven focus group discussions were conducted with 67 men. Men were willing to support their wives for screening and their daughters for HPV vaccination. Misperceptions such as family planning and poor personal hygiene leading to cervical cancer, and misperception of the preventative aspect of screening and vaccination were common. Women with cervical cancer suffer from stigmatization and family problems due to loss of fertility, less marital sexual activity, domestic violence and decreased economic productivity.

this study, because it contains verbatim transcriptions of FGDs with potentially sensitive information, including topics like HIV, STIs and extramarital sex. Researchers must provide their name, contact details, affiliation and objectives when requesting to review the original transcripts. All data and related metadata underlying the reported findings are available upon request from Murora Eva (Coordinator, African Rural University Research Unit) via email (emurora@aru.ac.ug) or telephone (+256783603938), and from Kasemiire Dorothy (Executive Assistant-Vice Chancellor's Office) via email (kdorothy@aru.ac.ug) or telephone (+256760109091), for researchers who meet the criteria for access of confidential data.

**Funding:** This research was supported by funding of Dioraphte Foundation (https://www.dioraphte.nl/), a non-commercial foundation from The Netherlands, granted to MF and JB via the Female Cancer Foundation (https://www.femalecancerfoundation.org/). The funders had no role in study design, data collection and analysis, decision to publish, or preparation of the manuscript. There was no additional external funding received for this study.

**Competing interests:** The authors have declared that no competing interests exist.

## Conclusions

Ugandan men were willing to support cervical cancer prevention for their wives and daughters after being informed about cervical cancer. Limited knowledge among men about the risk factors and causes of cervical cancer, and about the preventative aspect of HPV vaccination and screening and their respective target groups, can limit uptake of both services. Screening and vaccination programs should actively involve men in creating awareness to increase uptake and acceptance of prevention.

## Introduction

Although evidence-based preventive strategies for cervical cancer in low-resource setting have been developed, implementation and uptake of services remains low [1,2]. Prevention programs successfully avert cervical cancer by vaccination of young girls against human papillomavirus (HPV) and screening women for cervical intra-epithelial neoplasia (CIN lesions) which can eventually evolve into cervical cancer [3,4]. Successful implementation of national screening and vaccination programs in low- and middle-income countries (LMICs) is hampered by challenged health care systems, including lack of trained health workers, lack of equipment and inadequate sensitization of the community about these programs [5]. As a result, many women and girls do not have access to screening and vaccination services.

The burden of cervical cancer has an uneven distribution around the world, with over 80% of the cases occurring in LMICs [6]. In a country like Uganda, cervical cancer is the most common cancer in women during their reproductive age, and over 80% presents with advanced stage of disease [7]. Uganda developed a national Strategic Plan for Cervical Cancer and Control in 2010 including screening and treatment in a single visit and a national HPV vaccination program for girls from the age of ten [8,9]. At this stage, however, coverage of both vaccination and screening is low [5,7,10].

Several studies in Uganda analyzed women's perceptions about cervical cancer and screening. Common reasons not to attend screening were costs for transportation and screening services, distance to screening sites, lack of knowledge, fear for male health workers and fear for their partners opinion [11–13]. Programs that aim to address these barriers, often focus on women only, while little is known about the knowledge and beliefs of their partners or fathers.

Former qualitative research indicates that decision-making in health care is often related to gender inequality and male involvement should be incorporated into new interventions [10,14–16]. The World Health Organization (WHO) recommends to involve family members, in particular male partners, in health education [17]. However, studies about male perspectives on reproductive health services in Africa are rare. Mutyaba et al. showed the positive effect of involving male partners in follow-up of colposcopy in Kampala, with an 18% decrease in loss-to-follow up [18]. Muhindo et al. demonstrated the benefits of couple counseling in HIV/AIDS programs in Uganda, as more partners tested sero-negative on HIV and linkage to health care and treatment improved [19]. In a qualitative study about male perspectives on cervical cancer in Ghana, Williams et al. demonstrated that men's knowledge of cervical cancer was limited and men experienced barriers to support screening of their wives, such as disapproval of examination by male doctors and not recognizing benefits of preventive health care [20].

This study assessed male perspectives on cervical cancer prevention in Uganda to understand men's role in the decision-making process regarding the screening of their partners and

the HPV vaccination of their daughters. Increased understanding of the perspectives of men will support tailoring health education about cervical cancer prevention for men and encourage active male involvement in existing prevention programs resulting into increased uptake of preventative services.

# Materials and methods

## Design, setting and participants

A qualitative study with focus group discussions (FGDs) was conducted to assess male perspectives on cervical cancer, cervical cancer screening and HPV vaccination. The participants were men aged 25 to 60 years, who were married and/or had daughters. The study took place between June and August 2018, parallel to a screening program of the Uganda Rural Development and Training program (URDT), in Kagadi district, Mid-Western Uganda. The national HPV vaccination program in Uganda was rolled out in 2012 after an initial pilot phase in selected districts of Uganda. Kagadi district was not included in the pilot phase.

Kagadi district is a rural area consisting of 16 subcounties with a total population of 352,815 people [district data].

Simple random selection was used to select an urban, suburban and rural subcounty: Muhorro, Mabaale and Bwikara subcounty respectively.

Differences in age and social status could withhold participants from voicing their opinions. Younger men will not speak freely in the presence of older men, out of respect for elders. Therefore, FGD participants were purposely selected from different age groups: 25–35, 36–45, 46–60 years, and one group was composed of local leaders and other influential men in the community. For each FGD 6 to 8 participants were invited. The venues were located at a central and secluded place in each subcounty.

Participants were selected with simple and systematic sampling to create a diverse and representative group of the community. In each selected subcounty different parishes were randomly selected, followed by random selection of households on a list provided by the subcounty management. The selected households were visited by women involved in a rural transformation program of URDT in that specific subcounty and invited the male head of the household to participate in this study. The women were trained by two researchers [YS, BN] on the study protocol, participant selection and informed consent.

The FGD moderators [IS, JT] were two male lecturers from the African Rural University (ARU) in Kagadi with a local background, but without any close relationship with the participants or influential role in the subcounties. The moderators had experience in conducting qualitative research, were familiar with the local context and were fluent in Runyoro, the commonly spoken language in Kagadi district. The moderators received additional training by two researchers [YS, BN] in conducting FGDs, the study topic, the study protocol and informed consent.

The moderators guided the FGDs using the focus group guide [S1 File]. The themes for the focus group guide were based on literature search and findings of a baseline survey conducted by ARU in December 2017 before the start of the screening program [14,18,20,21]. The themes included female reproductive health, knowledge and beliefs on cervical cancer, cervical cancer screening, HPV vaccination and acceptance of screening and HPV vaccination. Following the theme 'knowledge and beliefs on cervical cancer', a short education was given on cervical cancer prevention to ensure all participants could formulate an opinion on the screening procedure and on HPV vaccination. The focus group guide was tested on a pilot group to assess acceptability and correct understanding of the questions. In an iterative process themes were further specified and extended over the course of the research. In addition, data analysis took

place directly after the FGDs to identify themes that needed further exploration in the following FGDs.

Prior to the FGD the moderators recorded characteristics of all participants (age, educational level, number of daughters, family affected by cervical cancer, current uptake in family of cervical cancer screening and HPV vaccination). All discussions were conducted in Runyoro. During the FGD a researcher was present to take notes about the context and observations during the discussion. At the end of each session the participants were asked to vote in confidentiality if they would accept screening for their wives.

## Data collection and analysis

All FGDs were audio recorded. The audio recordings were transcribed verbatim and translated from Runyoro into English by the moderators. The two moderators crosschecked the translations and checked if the translations captured all discussions that were recorded. Two independent researchers reviewed the transcripts and analyzed them inductively using thematic analysis without pre-identified codes [YS, MF]. ATLAS.ti version 8.2.31 for qualitative data management and MS Office Excel were used. The audio recordings were deleted after completing the analysis to ensure anonymity of the participating men.

Initially twelve FGDs were planned, however, this could be adjusted to achieve saturation of themes.

## Ethical considerations

The study protocol was reviewed and approved by the Uganda Cancer Institute Research and Ethics Committee (registration number 06–2018) and the Uganda National Council for Science and Technology (registration number SS4681). We obtained permission from Kagadi district and local leaders to access their communities and conduct this research. All participants provided written informed consent, in English and Runyoro, before the discussion and audio recording started. All participants were informed that they could withdraw from the discussion at any point. The participants were offered refreshments during the FGDs and were financially compensated for travel costs.

## Results

### Participant characteristics

In total eleven FGDs of approximately 1.5 hours each were conducted with 67 men. After nine FGDs saturation of themes was achieved. Two more FGDs were conducted to confirm saturation of themes and execute in-depth exploration of some subthemes. We noticed active engagement of all participants with good interaction between the participants themselves, and between the participants and the moderators.

Table 1 displays the characteristics of the participants. All participants were literate, and in 6 of the focus groups the majority had completed secondary or higher education. All participants were married, three participants (4%) stated that they had a relative with cervical cancer.

Table 2 illustrates the reported uptake of screening and HPV vaccination by participants, and acceptance of cervical cancer screening after the FGD. Uptake of preventive measures was low, 20% (11/56) of the men stated that their daughters were vaccinated and 16% (11/67) that their wives were screened. Directly following the FGD and briefing about cervical cancer and its prevention, 96% (64/67) of participants would accept their wives to be screened, one participant did not vote. These quantitative results were not analyzed further as the study was qualitative in design. However, the data gave insight in willingness to screen and actual uptake of screening.

**Table 1. Characteristics of FGD participants presented for each FGD.**

| FGD | Subcounty | Group | Number of participants | Median age (years) | Secondary or higher education | Median number of daughters |
|---|---|---|---|---|---|---|
| Pilot | Kagadi | 36–45 years | 6 | 36 | 83.3% | 2 |
| 1 | Bwikara | 46–60 years | 5 | 60 | 60% | 5 |
| 2 | Bwikara | Local leaders | 8 | 43 | 62.5% | 4 |
| 3 | Bwikara | 25–35 years | 6 | 27 | 100% | 2 |
| 4 | Bwikara | 36–45 years | 6 | 38 | 83.3% | 3 |
| 5 | Muhorro | Local leaders | 6 | 41.5 | 83.3% | 4 |
| 6 | Mabaale | Local leaders | 6 | 42 | 50% | 3 |
| 7 | Mabaale | 25–35 years | 6 | 32.5 | 16.7% | 2 |
| 8 | Mabaale | 36–45 years | 6 | 43.5 | 16.7% | 4 |
| 9 | Muhorro | 46–60 years | 6 | 46.5 | 33.3% | 4 |
| 10 | Muhorro | 25–35 years | 6 | 29 | 100% | 1 |

## Reproductive health and decision making

To understand the men's position in discussions on female reproductive health, men were asked about topics they have discussed with their wives. Men stated they talk with their wives about different reproductive health issues, including family planning and sexually transmitted infections (STIs). Some men only discussed STIs when signs and symptoms became more serious and treatment was indicated.

Men thought that women hide diseases from them out of fear of being suspected of extra-marital sexual activities and causing conflicts between husband and wife. Local leaders stated that, within their professional role, they also talk about female reproductive health with women from the community. In contrast, men are not supposed to talk to their daughters about reproductive health. Only mothers or close female relatives can discuss reproductive

**Table 2. Outcome of confidential poll on acceptance of screening and overview of current uptake of screening and vaccination presented for each FGD.**

| FGD | Participants (n[a]) | Acceptance for screening (n[a]) | Participants with daughters vaccinated (n[a]) | Participants with daughters screened (n[a]) | Participants with wife/wives screened (n[a]) |
|---|---|---|---|---|---|
| Pilot | 6 | 6 | 0 | 0 | 0 |
| 1 | 5 | 5 | 0 | 0 | 0 |
| 2 | 8 | 7 (1 did not vote) | 3 | 0 | 1 |
| 3 | 6 | 6 | 0 | 0 | 2 |
| 4 | 6 | 5 | 4 | 0 | 0 |
| 5 | 6 | 6 | 0 | 0 | 1 |
| 6 | 6 | 6 | 0 | 0 | 3 |
| 7 | 6 | 6 | 0 | 0 | 1 |
| 8 | 6 | 6 | 4 | 2 | 3 |
| 9 | 6 | 6 | 0 | 0 | 0 |
| 10 | 6 | 5 | 0 | 0 | 0 |

[a] n = number.

health with their daughters, because it is considered shameful for men. Men who want to inform their daughters on these subjects would ask the mothers to discuss reproductive health issues.

*"In our culture as 'Bunyoro' people, those issues a girl only discusses with the mother. A man can never enter into such discussions." (FGD 4)*

In the healthcare seeking process in Uganda the men's roles are prominent, because men provide money for transport and treatment. There were different views on who the final decision maker is in seeking healthcare, varying from wife and husband deciding together to the wife or husband deciding alone. These different responses were not linked to age differences, although participants stated that this concept has changed over time and men used to make the decisions for their family in the past.

## Causes and symptoms of cervical cancer

Most participants indicated that they had heard about cervical cancer before. However, some men stated the disease was new to them and had not been affecting their communities in the past. In most groups, men agreed that cervical cancer is a fatal disease, and advanced stages of disease are incurable due to unavailability of adequate treatment in Uganda or limited financial means to support expensive treatment. Participants thought that treatment in an early stage of the disease increases chances of survival.

*"When I understand that my woman is screened and she has cervical cancer, I feel very sad because I have a thinking that it is not easy to treat cancer, and this would be her beginning of death." (FGD 4)*

During the FGDs men stated that they had no idea about the causes and transmission. However, several men mentioned that cervical cancer is related to sexual intercourse and named some of the risk factors such as multiple sex partners, STIs, delivery at early age and uncircumcised men.

Compared to STIs, HIV infection was less frequently mentioned as a risk factor for cervical cancer. One local leader identified HPV infection through sexual contact as the major cause of cervical cancer.

Men held several misperceptions about the causes of cervical cancer. Recurring themes were family planning and poor hygiene. Participants stated that family planning can lead to irregular bleeding, associating this side-effect of family planning with a well-known symptom of cervical cancer. Another explanation given were 'coils' (intra-uterine device) and condoms causing wounds and consequently leading to infections that cause cervical cancer.

"*The water in the condoms can burn the inside parts or reproductive parts of women and then cause cervical cancer." (FGD 7)*

Many participants believed that traumatic events to the cervix could cause wounds in the cervix and therefore increase the risk of cervical cancer. The most common example was 'trauma through sexual intercourse', especially for young women.

"*For me, I think when younger girls get involved in sexual acts when their bodies are not yet mature, they can get damaged down there and hence {they develop} cervical cancer." (FGD 10)*

Poor male and female personal hygiene was a much-debated subject in every FGD. Participants agreed on 'uncircumcised men' as unhygienic, being a risk factor to transmit cervical cancer to women, and on circumcision as a very important preventive measure. Poor female hygiene was explained by 'too little bathing', 'wearing or sharing infected underwear' or 'urinating in infected latrines or toilets'.

*"If a woman has this cancer and she urinates somewhere and another woman also urinates on that same place, she is also likely to be affected with that cancer." (FGD 4)*

Poor hygiene during medical procedures in health facilities, like abortion, or during deliveries in the village were also mentioned as a possible cause of infection and leading to cervical cancer. Some men stated that use of sanitary pads during menstruation could increase chances of getting cervical cancer.

In all focus groups several symptoms of cervical cancer were mentioned, including pain, loss of libido and bleeding. In about one third of FGDs smell, weakness, weight loss and infertility were also identified as symptoms. Loss of libido for either the woman due to pain and weakness, or for the husband due to fear of infection and the affected uterus, was recognized by men in all focus groups and thought to possibly result into less intra-marital sex and driving men to find extramarital sex. This in turn could lead to family break-up and domestic violence. In a few groups miscarriage was mentioned as a possible symptom, due to insufficient closure of the cervix to a gravid uterus.

## Psychosocial consequences

In all discussions, men named severe distress and worries for both men and women about dying from cervical cancer. Some men said that cancer is worse than HIV/AIDS, because to their knowledge there is no medicine to treat cancer, unlike the availability of antiretroviral treatment for HIV/AIDS.

*"Many women are stigmatized because they know that cervical cancer can't be cured, so they know they will die soon." (FGD 6)*

*"Cancer scares men more than HIV/AIDS. AIDS has tablets, but with cancer, where are they? People with cancer, you will find a woman without her breast." (FGD 3)*

Some men stated that women might suspect cervical cancer to be a result of a co-wife's witchcraft. They also suggested that women would fear their husbands to divorce them, as he might suspect adultery as the cause of cervical cancer or be worried about female infertility as a consequence of cervical cancer. Additional consequences mentioned were family problems, such as domestic violence, and less economic productivity by the woman and therefore providing less support to her family and children.

*"This infection is through sexual intercourse so the man will know that his wife cheated on him, that is why she has cervical cancer (..) Now the man will start doubting his wife and he may chase her from his home." (FGD 10)*

Nearly everyone thought the community would find out about screening results and most participants mentioned fear of women for stigmatization by the community. The stigma arises from having a gynecological disease or having the uterus removed as treatment of cervical cancer.

*"A woman without uterus is no longer a woman." (FGD 3)*

Some men stated that the community could play a positive role for these women, offering financial support or counseling.

## Cervical cancer prevention

In all focus groups men stated that cervical cancer can be prevented, most importantly by sensitization of communities and avoiding the risk factors discussed in the previous paragraph. Preventable risk factors mentioned were male circumcision and avoiding 'irresponsible sex', whereby irresponsible sex was explained as multiple sex partners and extramarital sex.

*"I would like to tell them {other men} to protect themselves and stop having multiple sexual partners. They should also help to advise their wives not to cheat with other men." (FGD 2)*

Participants in most FGDs knew that screening and early treatment prevents cervical cancer, although this was not brought up at first. Vaccination was identified in only a few discussions as a preventive measure.

## Screening procedure and barriers for screening

Most men had no problem with the procedure of pelvic examination but did not like the idea of a male provider. In about half of the groups, men stated that expertise was more important than gender, but they still preferred a female provider. Men feared that male providers would want to have sexual intercourse with their wives, and some said they would want to be in the examination room if a male health worker was providing screening. Another reason mentioned for female preference was women feeling shy to be seen by male providers.

*"When she finds a female doctor screening, it becomes easy for her and then she tells her fellow women, they come running very fast for screening." (FGD 7)*

The most common barriers for women to attend screening was fear for pain and 'not seeing benefits in screening when there are no symptoms present'.

*"They say the instruments used during screening heavily affect women with a lot of pain and this generates fear into most women." (FGD 4)*

## HPV vaccination of daughters

About half of the men stated that they had heard about HPV vaccination, mostly through campaigns and awareness raising in schools, some men via radio or in church. A smaller group had never talked about HPV vaccination and clarified that their daughters were still too young to discuss this subject, although not all men were aware of the target age group for HPV vaccination.

In more than half of the focus groups men thought that the parents should decide together about vaccination of their daughters. Other men stated that it should be the mother or the head teacher in school to decide. Several groups requested for more information on HPV vaccination in the future and asked for immunization in schools.

The most important barrier for HPV vaccination was fear for infertility.

"*Others say that the plan is that doctors want to vaccinate our girls, daughters and end their productivity. That is why some parents do not want to vaccinate and circumcise their children, because many people are saying that they want women to have few children, so we need awareness*" (FGD 10)

When actively asked about fear of promiscuity as a result of HPV vaccination, men stated this would not be a major concern for them if girls are educated that HPV vaccination does not protect against other STIs.

It was evident there was confusion on the difference between HPV vaccination, cervical cancer screening and cervical cancer treatment. This could create barriers for both preventive strategies as illustrated below.

"*But village people have a lot of fear. Do they vaccinate the victims of cervical cancer or those who are not yet victims?*" (FGD 7)

## Awareness on cervical cancer and prevention

The radio was the most important source of information about cervical cancer and prevention. Other sources mentioned were 'women telling other women' and 'seminars and meetings'. Although a brief explanation about cervical cancer was given during the focus groups, many participants still had questions about family planning and hygiene in relation to cervical cancer. Everyone indicated that other community members should be informed on cervical cancer prevention. Most men felt elected to do so and requested for awareness materials as exemplified below.

"*I will not be different from my colleague. For me, I will be a speaker who will be a voice to move this message to my fellow men because we should not be silent. I will first talk to my family about today's meeting.*" (FGD 6)

When asked which media could be used for future awareness, several examples were given but most important media mentioned were radio, places of worship (such as church, mosque etc.), visual information and awareness at village level. Finally, when asked what message they wanted to share with other men in the community, all responded that husbands should support their women for screening.

"*We encourage them to use all means and take their wives for screening and treatment*" (FGD 9)

## Discussion

Our qualitative study using FGDs in Uganda demonstrated that men are supportive towards cervical cancer prevention for their wives and daughters. We identified multiple misconceptions about the causes of cervical cancer, and we found that cervical cancer leads to severe distress for families due to fear for a fatal outcome of the disease, stigmatization, and domestic violence.

To our knowledge this is the first qualitative study focusing on the perspectives of men regarding cervical cancer and its prevention in Uganda, and one of the few studies in LMICs looking at male perspectives. Men have proven to positively influence health seeking behavior

of women in HIV care, antenatal counseling, and follow-up of cervical cancer screening with colposcopy [14,15,18,19]. This study aimed to enhance male involvement in future awareness and preventive services for cervical cancer in Uganda and LMICs in general.

Mutyaba et al. reported that women in Uganda were under the assumption that their husbands were not interested in discussing female reproductive health or support them financially in seeking healthcare [22]. Yet, our study showed that men actually want to be involved in female healthcare in general and are willing to support women for cervical cancer screening and their daughters for HPV vaccination. This confirms the suggestion of Mutyaba et al. that men would be potentially willing partners if they are well informed [22].

In line with our findings, interviews in Ghana with male partners of healthy women in the community demonstrated that they would be willing to provide social, emotional, financial or material support if their wife would be diagnosed with cervical cancer [23]. However, women in that same community diagnosed with cervical cancer reported they did not always receive support from their husbands and some were abandoned due to the high costs of treatment or relationships with other women.

Male involvement and support are important factors for women to participate in screening programs and adhere to treatment and follow-up recommendations, as demonstrated in qualitative studies in Malawi and Kenya among women attending cervical cancer screening programs [24,25]. Women who did not adhere to the follow-up recommendations mentioned lack of financial and emotional support by their male partners and did not inform their male partners about their HPV-positive result.

A study among rural women in Tanzania demonstrated a significantly higher uptake of screening among women who perceived support from their husband compared to women who did not feel supported [16]. However, the distance to the screening facility and knowledge of cervical cancer and its prevention were the most important factors for uptake of screening.

In our study men recognized that they lacked knowledge on cervical cancer and its prevention and requested for better awareness in the community. These findings are similar to the results of studies in Ghana, Kenya and Uganda on the knowledge and beliefs of men about cervical cancer screening. Men reported lack of knowledge and indicated that increased knowledge could make them more supportive towards preventative programs [20,23,26–28]. In Kampala, Uganda, 65.1% of the male participants in a pre-education survey was not aware of the causes of cervical cancer and only 24.6% had heard about HPV. On the contrary, Mwaka et al. demonstrated good levels of awareness of cervical cancer and its risk factors and causes among both men and women in rural Northern Uganda [29]. However, although 70.3% of participants mentioned that cervical cancer is preventable, only 8.3% of participants thought that HPV vaccination could prevent cervical cancer and only 41.0% knew Pap smear screening as prevention method. The actual uptake of screening and HPV vaccination among the study population was not specified.

Misperceptions can influence healthcare seeking behavior. Turiho et al. reported that rumors and concerns about HPV vaccination among parents in Uganda reduced the acceptance of vaccination for their daughters [9]. Over time, the perceived benefits enhanced the acceptability of HPV vaccination by the parents. In India Basu et al. found that targeted education of parents led to increased acceptance of HPV vaccination by both male and female parents after initial refusal of vaccination [30]. A similar effect of health education has been demonstrated for the acceptance of screening. A survey conducted among men in Kampala showed a high acceptance of screening and vaccination, after receiving one health education session about cervical cancer prevention [28].

Another barrier to uptake of screening is the fear for cervical cancer, which is perceived as a serious and potentially fatal disease without available treatment. Studies in Kenya and Uganda

also reported that the severe outcome of cervical cancer could be a potential motivation to participate in prevention programs, contradictorily, fear for the screening results and the consequences of cervical cancer was also identified as a barrier for screening [9,26]. Misperceptions about the screening result can further withhold men and women from taking the right preventive measures. For example, in our study a positive result in screening was confused with invasive cervical cancer. This could lead to stigmatization of women attending screening services and consequently less uptake of screening.

In previous studies on women's attitudes towards screening in Uganda, Tanzania and Bangladesh, embarrassment about the screening procedure and gender of the provider were identified as barriers to attend screening [10,12,16,31]. Gynecological examination and the gender of the provider also emerged as sensitive subjects in our study. Many men distrust male providers in conducting a gynecological examination, although the importance of expertise above gender of the health care provider was pointed out. Future screening services should therefore focus on creating a safe environment and making health workers aware of these gender-related barriers.

Although previous studies illustrated that urbanization levels could influence knowledge of participants, and consequently attitudes towards screening, we did not find major differences in knowledge between urban and rural areas [10]. This could be explained by the relatively small differences between urban, suburban and rural areas in Kagadi district, compared to Masaka district where the study of Twinomujuni et al. was conducted. Furthermore, all participants were literate, and most men had finished primary or secondary school. This is similar to the national figures, with 83% of the male population being literate [32].

Our findings about stigmatization and discrimination, related to infertility, extramarital sexual activity and less economic productivity are in line with a systematic review about barriers to uptake of screening in Sub Sahara Africa [33]. However, men also mentioned the supportive role of the community, either financially or by counseling. Future studies could explore this role and the effect of programs directed to community support.

We recommend that awareness programs should address the importance of prevention for all women and girls, and at the same time invest in thorough health education for both women and men at community level. Health education should include information about misconceptions related to family planning and personal hygiene, inform that early detection and treatment can prevent fatal outcome and provide contextualized information about the preventative aspect of vaccination and screening, including the importance of male support for preventive programs. Men should be encouraged to actively engage in health education activities, to support their wives and daughters to participate in screening and vaccination programs, and to provide financial support if needed. In Uganda, radio and community conversations can be effective communication channels to reach men, and men should be included in the development of visual health education materials.

## Limitations

This study was conducted in collaboration with ARU and URDT, both local organizations working on rural development, with years of experience and knowledge of the communities in Kagadi District. Therefore, ARU and URDT were able to guide in selection of participants, moderating group discussions, and interpretation of results. By organizing groups in different age categories, social status, and local background we believe we have been able to create a diverse and representative sample of the male community in Kagadi. Nonetheless, we are aware that socially desired answers cannot be ruled out and that they can mask taboos. For example, the large acceptance of screening might not be reflected by screening of the

participants' wives in practice. We recommend to actively approach partners of women that did not participate in available screening programs for in-depth interviews. We further recommend to approach couples after participation in screening programs to learn more about the level of support women experience from their partners and how this influenced their decision to participate. In practice, however, it might be difficult to obtain consent from non-cooperative partners and partners of non-attenders.

Another limitation of our study is translation bias. All transcripts were translated from Runyoro to English. The translators are both native Runyoro speaking and fluent in English, as this is the official language in Uganda. To prevent translation bias, all English transcripts were cross-checked, and all text was interpreted by local researchers. Despite these measures, there is a risk of translation bias or missing context-specific information.

This study was conducted in a rural area in the Mid-Western region of Uganda with a diverse population. Our results cannot directly be applied to different communities in Uganda or LMICs in general.

## Conclusions

This study is one of the first qualitative studies about the perspectives of men on cervical cancer prevention in Africa. We found men were willing to support their wife and daughters in preventative measures, and at the same time identified several misconceptions about the preventative aspect of screening and vaccination and the risk factors and causes of cervical cancer. Cervical cancer burdens women, men and their families, alongside less economic productivity, stigmatization, and domestic violence. Current awareness programs about primary and secondary prevention focus on women. Our findings are encouraging to actively involve men and their communities in cervical cancer prevention programs in order to increase acceptance and achieve nationwide uptake of services.

## Supporting information

**S1 File. Focus group discussion guide.**
(DOCX)

**S2 File.**
(DOCX)

## Acknowledgments

The authors would like to express their gratitude to all men that participated in the focus groups and willingly shared their opinions and beliefs with us. Our thanks and appreciation go out to the staff of the Ugandan Rural Development and Training Program, in particular the screening team, and African Rural University for their support for this research project.

## Author Contributions

**Conceptualization:** Marlieke de Fouw, Ben Niwagaba, Jogchum Jan Beltman.

**Formal analysis:** Marlieke de Fouw, Yaël Stroeken, Ria Reis.

**Investigation:** Yaël Stroeken, Ben Niwagaba, John Tusiime, Isingoma Sadayo.

**Methodology:** Marlieke de Fouw, Yaël Stroeken, Ben Niwagaba, Ria Reis, Alexander Arnold Willem Peters, Jogchum Jan Beltman.

**Project administration:** Yaël Stroeken.

**Supervision:** Marlieke de Fouw.

**Visualization:** Marlieke de Fouw.

**Writing – original draft:** Marlieke de Fouw.

**Writing – review & editing:** Marlieke de Fouw, Yaël Stroeken, Ben Niwagaba, Mwalimu Musheshe, John Tusiime, Isingoma Sadayo, Ria Reis, Alexander Arnold Willem Peters, Jogchum Jan Beltman.

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
