## [Decision Letter · Decision Letter 0]

9 Mar 2022

PONE-D-22-00308Involving men in cervical cancer prevention; a qualitative enquiry into male perspectives on screening and HPV vaccination in Mid-Western UgandaPLOS ONE

Dear Dr. de Fouw,

Thank you for submitting your manuscript to PLOS ONE. After careful consideration, we feel that it has merit but does not fully meet PLOS ONE’s publication criteria as it currently stands. Therefore, we invite you to submit a revised version of the manuscript that addresses the points raised during the review process.

We look forward to receiving your revised manuscript.

Kind regards,

Anne F. Rositch, PhD, MSPH

Academic Editor

PLOS ONE

Journal Requirements:

3. Our staff editors have determined that your manuscript is likely within the scope of our Cancer and Social Inequity Call for Papers. This editorial initiative is headed by a team of Guest Editors for PLOS ONE: Vesna Zadnik (Institute of Oncology, Ljubljana), Nixon Niyonzima (Uganda Cancer Institute), Claudia Allemani (London School of Hygiene and Tropical Medicine). This call for papers aims to highlight the negative impacts of social inequities on health, identify the effects of social and corporate policies on access to healthcare services, and propose solutions to promote more equitable cancer outcomes and ultimately, social justice.  Additional information can be found on our announcement page: https://collections.plos.org/call-for-papers/cancer-and-social-inequality/

If you would like your manuscript to be considered for this collection, please let us know in your cover letter and we will ensure that your paper is treated as if you were responding to this call.  Please note that being considered for the Collection does not require additional peer review beyond the journal’s standard process and will not delay the publication of your manuscript if it is accepted by PLOS ONE. If you would prefer to remove your manuscript from collection consideration, please specify this in the cover letter.

(This research was supported by funding of Dioraphte Foundation from The Netherlands. The funders had no role in study design, data collection and analysis, decision to publish, or preparation of the manuscript.)

Reviewers' comments:

Reviewer's Responses to Questions

**Comments to the Author**

1. Is the manuscript technically sound, and do the data support the conclusions?

Reviewer #1: Yes

Reviewer #2: Partly

2. Has the statistical analysis been performed appropriately and rigorously? 

Reviewer #1: N/A

Reviewer #2: N/A

3. Have the authors made all data underlying the findings in their manuscript fully available?

Reviewer #1: No

Reviewer #2: Yes

4. Is the manuscript presented in an intelligible fashion and written in standard English?

Reviewer #1: Yes

Reviewer #2: Yes

5. Review Comments to the Author

Reviewer #1: This is important work addressing a critical barrier to women's engagement in cervical cancer screening in Uganda and I'm thrilled to see this work is happening. I would suggest condensing Figure 1 and its current format is more of a distraction than beneficial to the background of the paper. In the results section the quotes chosen for specific themes did not always seem intuitive and sometimes even detracted from what was being emphasized, perhaps the theme itself was a bit of a stretch or a different quotation could be used to ensure a more coherent message.

Finally there has been other work in Uganda looking at men's attitudes to screening including the ASPIRE group (Understanding Men’s Perceptions of Human Papillomavirus and Cervical Cancer Screening in Kampala, Uganda, Moses et al 2018) which would be a good reference to add.

Reviewer #2: The study presents relevant insights on the need to engage the male gender in improving health literacy and potentially practice of cancer prevention strategies for cervical cancer. Their key role within the household is often overlooked and this presents a unique opportunity to improve knowledge, communication and uptake of screening and vaccination while also demystifying cervical cancer and reduce stigma. The other strong component of this paper are the opportunities for intervention from the documented false perceptions identified within the communities.

I have a few questions listed thus: As documented in the limitation - "We are aware that socially desired answers cannot be ruled out and that they can mask taboos. The large acceptance of screening after the FGD might not be reflected by screening of their wives in practice" - Was there any suggestion from the FGDs that could mitigate against this threat? This would provide further opportunities that can be exploited to improve practice of screening. I'm concerned that the method of participant selection may have contributed to the suspicion of "socially desired answers" and this may not allow for further interrogation on the possible barriers to increasing uptake.

Overall, I think the study contributes to the body of knowledge by showcasing the need to study the local setting, identify key stakeholders and adapt interventions. It also describes relevant local challenges and opportunities for improving cancer detection and is reproducible, but may require more stringent methods of stakeholder engagement to avoid bias.

6. PLOS authors have the option to publish the peer review history of their article (what does this mean?). If published, this will include your full peer review and any attached files.

Reviewer #1: No

Reviewer #2: **Yes: **Nwamaka Lasebikan

---

## [Author Response · Author response to Decision Letter 0]

30 Aug 2022

Reviewer 1 I would suggest condensing Figure 1 and its current format is more of a distraction than beneficial to the background of the paper. 

Thank you for your suggestion to make the table more informative. We have removed the last column of the table (number of participants with a relative with cervical cancer) and the median in the column with number of daughters per relatives. We also combined two categories of educational level (secondary and higher education) in a percentage as this is most relevant.

Reviewer 1 In the results section the quotes chosen for specific themes did not always seem intuitive and sometimes even detracted from what was being emphasized, perhaps the theme itself was a bit of a stretch or a different quotation could be used to ensure a more coherent message 

Thank you for your comment about the quotations we selected. We have reviewed all quotations with the team of authors following your review and discussed about their relevance for each theme, it is one of the challenges in qualitative research. We decided to delete some quotes which could distract the reader and which was already sufficiently reflected in the text. For some themes the topics mentioned were indeed surprising. On the other hand, we used the themes in our FGD guide to structure the discussion but followed the topics and discussion the men raised during the FGD. Therefore the topics and quotes might sometimes not feel entirely coherent. Furthermore, the English grammar of some quotes is not entirely correct but we decided to keep the quotes as close to the original version as possible. We feel that the current quotes do reflect the perspective of men as expressed during the different FGDs.

Reviewer 1 Finally there has been other work in Uganda looking at men's attitudes to screening including the ASPIRE group (Understanding Men’s Perceptions of Human Papillomavirus and Cervical Cancer Screening in Kampala, Uganda, Moses et al 2018) which would be a good reference to add. 

Thank you for this literature suggestion, this manuscript is included as reference number 25.

Reviewer 2 As documented in the limitation - "We are aware that socially desired answers cannot be ruled out and that they can mask taboos. The large acceptance of screening after the FGD might not be reflected by screening of their wives in practice" - Was there any suggestion from the FGDs that could mitigate against this threat? This would provide further opportunities that can be exploited to improve practice of screening. I'm concerned that the method of participant selection may have contributed to the suspicion of "socially desired answers" and this may not allow for further interrogation on the possible barriers to increasing uptake. 

Thank you for your remark and suggestions for future research in this field. During the FGDs we noticed active engagement of all participants with good interaction between the participants themselves and the participants and the moderators. After the FGDs men indicated that they felt relieved they could talk about the subject and that we were seriously interested in their perspective. From our experience in the screening project in Kagadi we found that many men would want their wives to be screened and stay healthy, but lack of knowledge about cervical cancer and the impact of screening, and poor access to screening are important barriers for their support. We purposely selected different age groups because in this context younger men will not feel comfortable to speak freely out of respect for elders. In addition, we had a separate FGD with influential men in the community. 

In future research we would consider approaching husbands of women that did not participate in screening, where this has been actively offered to all women in the community, for in-depth interviews. Another alternative could be to ask women coming for screening if they felt supported by their men, and approach their men for in-depth interviews (from both women who felt supported and who felt not supported), however, this will be difficult in practice to obtain consent and to get in contact with them. 

In order to gain more insight, we hope our research shows that men like to be involved and therefore we advise to include men from the beginning in cervical cancer awareness and programming of prevention, and involve them actively in community engagement. This will also help to better understand their perspective and possible barriers and opportunities.

---

## [Editor Report · Decision Letter 1]

10 Oct 2022

PONE-D-22-00308R1Involving men in cervical cancer prevention; a qualitative enquiry into male perspectives on screening and HPV vaccination in Mid-Western UgandaPLOS ONE

Dear Dr. de Fouw,

Thank you for submitting your manuscript to PLOS ONE. After careful consideration, we feel that it has merit but does not fully meet PLOS ONE’s publication criteria as it currently stands. Therefore, we invite you to submit a revised version of the manuscript that addresses the points raised during the review process.

We look forward to receiving your revised manuscript.

Kind regards,

Anne F. Rositch, PhD, MSPH

Academic Editor

PLOS ONE

Journal Requirements:

Additional Editor Comments (if provided):

While the authors addressed the original reviewer comments in the response document, not all translated to revisions of the manuscript itself. Specifically, I would like to see the manuscript edited to reflect the question/discussion on the final reviewer item, especially since the information provided by the authors with regards to age-based selection and additional FGD's with influential participants isn't outlined/consistent with methods.

Reviewer 2 As documented in the limitation - "We are aware that socially desired answers cannot be ruled out and that they can mask taboos. The large acceptance of screening after the FGD might not be reflected by screening of their wives in practice" - Was there any suggestion from the FGDs that could mitigate against this threat? This would provide further opportunities that can be exploited to improve practice of screening. I'm concerned that the method of participant selection may have contributed to the suspicion of "socially desired answers" and this may not allow for further interrogation on the possible barriers to increasing uptake.

Thank you for your remark and suggestions for future research in this field. During the FGDs we noticed active engagement of all participants with good interaction between the participants themselves and the participants and the moderators. After the FGDs men indicated that they felt relieved they could talk about the subject and that we were seriously interested in their perspective. From our experience in the screening project in Kagadi we found that many men would want their wives to be screened and stay healthy, but lack of knowledge about cervical cancer and the impact of screening, and poor access to screening are important barriers for their support. We purposely selected different age groups because in this context younger men will not feel comfortable to speak freely out of respect for elders. In addition, we had a separate FGD with influential men in the community.

In future research we would consider approaching husbands of women that did not participate in screening, where this has been actively offered to all women in the community, for in-depth interviews. Another alternative could be to ask women coming for screening if they felt supported by their men, and approach their men for in-depth interviews (from both women who felt supported and who felt not supported), however, this will be difficult in practice to obtain consent and to get in contact with them.

In order to gain more insight, we hope our research shows that men like to be involved and therefore we advise to include men from the beginning in cervical cancer awareness and programming of prevention, and involve them actively in community engagement. This will also help to better understand their perspective and possible barriers and opportunities.
---

## [Author Response · Author response to Decision Letter 1]

24 Nov 2022

We reviewed our reference list and updated the references and their lay-out with track changes in our revised manuscript according to the PLOS ONE guidelines. We did not identify incorrect or retracted references.

In addition to our previous rebuttal letter, we would like to clarify our revisions:

1. The first reference of Gakidou et al, was replaced by Bruni et al, because this is the most recent publication about the coverage of cervical cancer screening in LMICs and this paper was not yet available at the time of our initial submission.

2. For reference 8 we included the updated link, to access the National Strategic Plan of Uganda.

3. The other revisions are the adjustment of the presentation of the journal titles to the uniform Vancouver system.

Thank you for your suggestion to edit our manuscript to bring these considerations to the attention or the reader, and not only the reviewers. We have adjusted the relevant sections in methods, results, and discussion in track changes.

---

## [Editor Report · Decision Letter 2]

20 Dec 2022

Involving men in cervical cancer prevention; a qualitative enquiry into male perspectives on screening and HPV vaccination in Mid-Western Uganda

PONE-D-22-00308R2

Dear Dr. de Fouw,

We’re pleased to inform you that your manuscript has been judged scientifically suitable for publication and will be formally accepted for publication once it meets all outstanding technical requirements.

Kind regards,

Anne F. Rositch, PhD, MSPH

Academic Editor

PLOS ONE
---

## [Editor Report · Acceptance letter]

20 Jan 2023

PONE-D-22-00308R2 

Involving men in cervical cancer prevention; a qualitative enquiry into male perspectives on screening and HPV vaccination in Mid-Western Uganda 

Dear Dr. de Fouw:

I'm pleased to inform you that your manuscript has been deemed suitable for publication in PLOS ONE. Congratulations! Your manuscript is now with our production department. 

Kind regards, 

on behalf of

Dr. Anne F. Rositch 

Academic Editor

PLOS ONE